# Using electronic health records to streamline provider recruitment for implementation science studies

**Chiamaka L. Okorie**[1], **Elise Gatsby**[2], **Florian R. Schroeck**[1,3,4,5,6], **A. Aziz Ould Ismail**[3], **Kristine E. Lynch**[2]*

**1** From Geisel School of Medicine at Dartmouth College, Lebanon, NH, United States of America, **2** VA Salt Lake City Health Care System and University of Utah, Salt Lake City, UT, United States of America, **3** White River Junction VA Medical Center, White River Junction, VT, United States of America, **4** Section of Urology Dartmouth Hitchcock Medical Center, Lebanon, NH, United States of America, **5** The Dartmouth Institute for Health Policy and Clinical Practice, Lebanon, NH, United States of America, **6** Norris Cotton Cancer Center Dartmouth Hitchcock Medical Center, Lebanon, NH, United States of America

* Kristine.Lynch@VA.gov

**Data Availability Statement:** Data cannot be shared publicly because they contain potentially identifying and sensitive patient information. Data

## Abstract

### Background

Healthcare providers are often targeted as research participants, especially for implementation science studies evaluating provider- or system-level issues. Frequently, provider eligibility is based on both provider and patient factors. Manual chart review and self-report are common provider screening strategies but require substantial time, effort, and resources. The automated use of electronic health record (EHR) data may streamline provider identification for implementation science research. Here, we describe an approach to provider screening for a Veterans Health Administration (VHA)-funded study focused on implementing risk-aligned surveillance for bladder cancer patients.

### Methods

Our goal was to identify providers at 6 pre-specified facilities who performed ≥10 surveillance cystoscopy procedures among bladder cancer patients in the 12 months prior to recruitment start on January 16, 2020, and who were currently practicing at 1 of 6 pre-specified facilities. Using VHA EHR data (using CPT, ICD10 procedure, and ICD10 diagnosis codes), we identified cystoscopy procedures performed after an initial bladder cancer diagnosis (i.e., surveillance procedures). Procedures were linked to VHA staff data to determine the provider of record, the number of cystoscopies they performed, and their current location of practice. To validate this approach, we performed a chart review of 105 procedures performed by a random sample of identified providers. The proportion of correctly identified procedures was calculated (Positive Predictive Value (PPV)), along with binomial 95% confidence intervals (CI).

### Findings

We identified 1,917,856 cystoscopies performed on 703,324 patients from October 1, 1999 —January 16, 2020, across the nationwide VHA. Of those procedures, 40% were done on

are available via the Veteran's IRB of Northern New England (contact via email: vhawrjresearchtask@va.gov) for researchers who meet the criteria for access to confidential Department of Veterans Affairs data. Data in the Department of Veterans Affairs Corporate Data Warehouse are collected for clinical purposes as part of the patient medical record. They contain potentially identifying and sensitive patient information and, therefore, cannot be shared. They can be accessed by any VA researcher through the Institutional Review Board process. Interested researchers can direct data access requests to the director of the Veteran's IRB of Northern New England, 215 N Main Street, White River Junction, VT 05009, phone 802-295-9363, email: vhawrjresearchtask@va.gov.

**Funding:** FRS is supported by the Department of Veterans Affairs Health Services Research & Development (IIR 18-215; I01HX002780-01). The funding organizations had no role in the design and conduct of the study; collection, management, analysis, and interpretation of the data; preparation, review, or approval of the manuscript; and decision to submit the manuscript for publication.

**Competing interests:** The authors have declared that no competing interests exist.

**Abbreviations:** CDW, Corporate Data Warehouse; CI, Confidence Interval; CPT, Common Procedural Terminology; HER, Electronic Health Records; ICD, International Classification of Diseases; ImpRaBS, Implementation of Risk-aligned Bladder Cancer Surveillance; OMOP, Observational Medical Outcomes Partnership; PPV, Positive Predictive Value; VA, Department of Veterans Affairs; VHA, Veterans' Health Administration; VINCI, VA Informatics and Computing Infrastructure.

patients who had a prior record of bladder cancer and were completed by 15,065 distinct providers. Of those, 61 performed $\geq$ 10 procedures and were currently practicing at 1 of the 6 facilities of interest in the 1 year prior to study recruitment. The random chart review of 7 providers found 101 of 105 procedures (PPV: 96%; 95% CI: 91% to 99%) were surveillance procedures and were performed by the selected provider on the recorded date.

## Implications

These results show that EHR data can be used for accurate identification of healthcare providers as research participants when inclusion criteria consist of both patient- (temporal relationship between diagnosis and procedure) and provider-level (frequency of procedure and location of current practice) factors. As administrative codes and provider identifiers are collected in most, if not all, EHRs for billing purposes this approach can be translated from provider recruitment in VHA to other healthcare systems. Implementation studies should consider this method of screening providers.

## Introduction

The goal of implementation science is to promote evidence-based findings into real-world practice [1,2]. Healthcare providers are often targeted as participants in implementation science research to gain insights into implementation barriers and help develop strategies to enhance uptake of guideline-recommended practices. Provider recruitment, although a single milestone to the research process, is comprised of multiple sub-components, each with its challenges [3]. One of the first steps of provider recruitment is screening, which is identifying individuals who meet defined eligibility criteria for study participation. Like patient recruitment strategies for randomized clinical trials, eligibility criteria are necessary to ensure homogeneity across important factors related to the research question. However, with provider recruitment, eligibility is frequently based on both provider and patient factors.

Historically, implementation research has relied on two screening strategies to identify potentially eligible providers: 1) self-report of clinical data and 2) manual chart review [4,5]. Although these methods can be effective, self-report hinges on the accuracy of provider recall [6], and manual review often takes longer than anticipated, which can adversely affect research timelines, funds, and processes [5]. Recent literature continues to call for alternative approaches that can help improve provider recruitment efficiency throughout the life cycle of an implementation study [7]. Electronic health records (EHRs) contain data that provide one such alternative approach [8–11]. Automated screening based on EHR data has been used for patient screening for randomized clinical trials recruitment [12]. However, an extension of this approach has not been described for provider screening and recruitment. Since provider data can be linked to patient data, the EHR also provides an avenue to find providers based on patient- and provider-level exclusion/inclusion criteria. Thus, the widespread adoption and use of EHRs may offer investigators the ability to rapidly identify a precise cohort of providers prescreened for eligibility criteria for participation in implementation studies [13–15].

In this manuscript, we provide an example of streamlined provider screening employing EHR data using the Veterans Health Administration (VHA) study focused on the Implementation of Risk-aligned Bladder Cancer Surveillance (ImpRaBS) as a use case [16]. The VHA is one of the largest integrated health systems in the United States and was one of the earliest

adopters of electronic records [17,18]. Data derived from its in-house EHR system, Veterans Health Information Systems and Technology Architecture (Vista), is available to researchers via the VHA Corporate Data Warehouse (CDW) [19]. The CDW is a nationwide data repository of prioritized clinical domains including demographic, laboratory, pharmacy, procedure, and vital status data, as well as unstructured clinical text [20]. It is refreshed nightly enabling near-real-time querying of data for clinical and administrative research purposes. For this project, this rich resource was leveraged to identify providers eligible for inclusion based on patient- and provider-level criteria.

## Methods

### Use case

The primary aim of ImpRaBS is to develop and subsequently pilot implementation strategies for risk-aligned bladder cancer surveillance [16]. Cystoscopy is a surgical procedure that allows a urologist to visualize abnormalities of the urethra and bladder. It is the most common surgical procedure in the VHA with 30,000 performed annually [21]. Using cystoscopy, bladder cancer patients undergo frequent surveillance to check for cancer recurrence within their bladder. There is international consensus that surveillance cystoscopy should be aligned with each patient's risk for recurrence and progression [22], however, prior work suggested that risk-aligned surveillance of patients with non-muscle invasive bladder cancer was not consistently practiced in VHA [16]. Thus, the goal of this study was to develop implementation strategies to improve risk-aligned surveillance of early-stage bladder cancer. As a first step towards this goal, we evaluated factors influencing guideline adherent clinical practice through provider interviews [23]. Thus, we needed to identify potentially eligible VHA providers for a qualitative evaluation of the barriers and facilitators of risk-aligned bladder cancer surveillance.

### Eligibility criteria

Inclusion criteria consisted of patient- and provider-level data extracted from the CDW. Providers were eligible for inclusion if they were currently practicing at 1 of 6 pre-specified VHA Medical Centers and had performed ≥ Medcystoscopy procedures among previously diagnosed bladder cancer patients (i.e., surveillance cystoscopy) in the 12 months prior to recruitment start (January 16, 2020). Attending urologists, residents, and Advance Practice Providers (NP/PA) were all considered for inclusion and considered currently practicing if at least one of their qualifying procedures was performed at one of the 6 study sites.

We identified eligible providers using four sequential steps (see Fig 1):

1. Identification of surveillance cystoscopy procedures using patient-level data.

2. Linkage of provider data (provider ID and current location) to these procedures.

3. Application of eligibility criteria to linked providers.

4. Validation of at least 100 procedures by manual chart review.

**Patient level.** We used patient-level data to identify cystoscopy procedures. Because administrative coding (i.e., procedure codes) is not specific to surveillance cystoscopy, we developed a simple rule-based phenotype that considered diagnosis and procedure data elements. Surveillance cystoscopy was defined as the occurrence of a cystoscopy procedure at any time after a given patient's incident bladder cancer diagnosis. Bladder cancer was defined based on the International Classification of Diseases, Ninth and Tenth Revision diagnosis codes (ICD-9, ICD-10). Cystoscopy was defined using Common Procedural Terminology

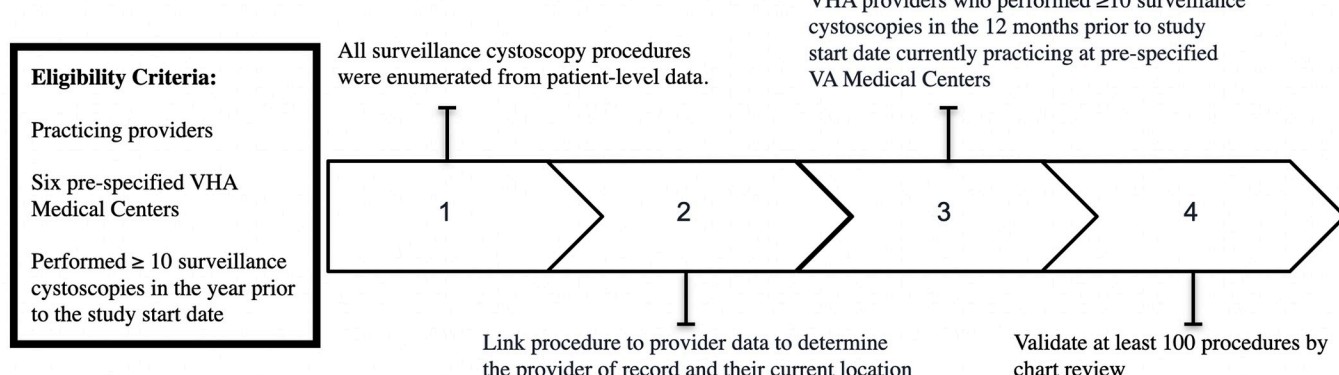

**Fig 1. Flow chart showing the sequential identification process of eligible providers based on inclusion and exclusion criteria.**

(CPT) codes and ICD-10 procedure codes (see S1 Appendix for diagnosis and procedure codes). Bladder cancer diagnoses and cystoscopy procedures from October 1, 1999—January 16, 2020, were considered.

**Provider level.** All surveillance cystoscopy procedures were enumerated from patient-level data as described above and were each linked to the provider and clinical facility of record. For the "provider" role, the study recruited urologists and PAs. At the VHA, because provider identifiers are specific to each medical facility, providers can theoretically have up to 130 distinct identifiers. We established a many-to-one relationship using the pre-transformed VHA data available in the Observational Medical Outcomes Partnership (OMOP) common data model, which permits only one record per provider [24].

For each provider, we then calculated the total number of surveillance procedures performed from October 1, 1999—January 16, 2020, regardless of the geographical location the procedure was performed. Any provider who performed ≥10 cystoscopy procedures in the one year prior to study participation and currently practicing at any of the 6 clinical locations of interest was retained (Fig 1). If any of the ≥10 cystoscopy procedures in the 12 months prior to recruitment start were performed at 1 of the 6 pre-specified VHA Medical Centers, that provider was categorized as currently practicing at that Medical Center. A list of all qualifying providers and up to 15 of their most recent surveillance cystoscopy procedures, including patient identifiers and the procedure CPT, ICD10 procedure, and ICD10 diagnosis codes, was generated for validation.

## Validation

Each provider from the list generated above was assigned a random number. The list was then sorted by the random numbers in ascending order. Starting with provider number 1, all individual procedures were reviewed chronologically per provider until reaching at least 100 procedures. Given that the first 7 randomly selected providers all had 15 procedures each, 105 procedures were reviewed.

Manual chart review by study personnel (a urologist and a study coordinator) served as the reference standard for validation. They manually validated bladder cancer surveillance procedures by referencing procedure notes on or around the date of cystoscopy and determined (1) if a cystoscopy was performed on the date indicated by the CDW data, (2) if it occurred after a bladder cancer diagnosis and thus was a surveillance procedure, (3) whether the provider who

performed the procedure was correctly identified, and (4) whether the location of the procedure aligned with the extracted data. The proportion of correctly identified procedures was calculated (Positive Predictive Value (PPV)), along with binomial 95% confidence intervals (CI).

### Ethics statement

The study was approved by the VA Central Institutional Review Board (No.19-01).

Data were not anonymized given the need to link patient and provider records and perform a chart review for validation. This study had a waiver of informed consent and HIPAA authorization related to the data and study activities described in the current manuscript.

## Results

### Patient level

Using the cystoscopy codes listed in the S1 Appendix, there was a total of 1,917,856 distinct cystoscopy procedures performed on 703,324 patients in VHA from October 1, 1999—January 16, 2020. During the same period, 850,305 cystoscopy procedures were performed on the 250,643 patients who had bladder cancer. Of those procedures, 762,158 met the definition of surveillance cystoscopy, that is the bladder cancer diagnosis preceded cystoscopy (circle section on the left side of Fig 2).

### Provider level

A total of 15,065 distinct providers performed the 762,158 procedures determined to be surveillance cystoscopy. Of these, 1,005 providers were located at the 6 pre-specified VHA Medical Centers. Of the 15,065 providers, 745 performed procedures in the 12 months prior to provider participation in study. At the intersection of location and number of surveillance cystoscopies performed in the prior year, there were only 61 providers who performed ≥10 procedures in that prior 12 months and were practicing at 1 of the 6 pre-specified facilities. These 61 providers were considered eligible for recruitment (Fig 2) with an average of 31.4 patients seen by these eligible providers and an average of 34.1 procedures performed. Below we

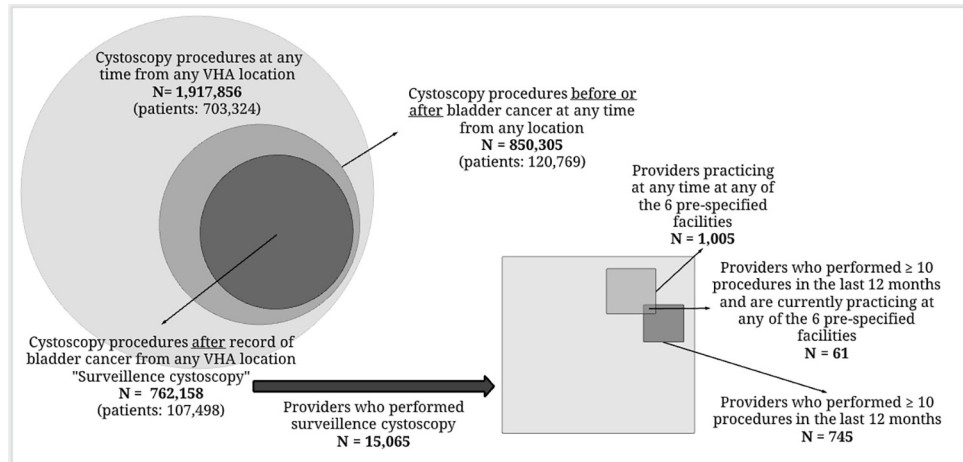

**Fig 2. Pictogram describing the stepwise descent from patient data to the final list of eligible providers who performed qualifying surveillance cystoscopy.**

**Table 1. Detailing the minimum, median and maximum number of procedures and unique patients for selected provider groups with averages and SD, including the specific procedure timeline.**

| | Providers who performed surveillance cystoscopy | | Providers practicing at any time at any of the 6 stations | | Providers who performed > = 10 procedures in the last 12 months | | Providers who performed > = 10 procedures in the last 12 months at the 6 stations | |
|---|---|---|---|---|---|---|---|---|
| | N = 15,065 | | N = 1,005 | | N = 745 | | N = 61 | |
| | Patients | Procedures | Patients | Procedures | Patients | Procedures | Patients | Procedures |
| Minimum | 1.0 | 1.0 | 1.0 | 1.0 | 7.0 | 10.0 | 9.0 | 10.0 |
| Median | 2.0 | 2.0 | 4.5 | 5.0 | 31.0 | 33.0 | 27.0 | 27.0 |
| Maximum | 1212.0 | 4205.0 | 495.0 | 1791.0 | 329.0 | 405.0 | 82.0 | 97.0 |
| Average | 28.9 | 46.9 | 30.8 | 42.7 | 42.4 | 51.1 | 31.4 | 34.1 |
| SD | 61.8 | 161.2 | 52.2 | 115.3 | 36.1 | 49.5 | 18.2 | 21.0 |
| | Procedures between 1999 and 01/16/2020 | | Procedures between 1999 and 01-16-2020 | | Procedures between 01/16/2019 and 01/16/2020 | | Procedures between 01/16/2019 and 01/16/2020 | |

provide a table detailing the number (median and interquartile range) of procedures and unique patients for each selected provider groups before final eligibility criteria were met (Table 1).

## Results of validation

105 cystoscopies performed by 7 providers according to the administrative data were reviewed. 101 of 105 surveillance procedures (PPV of 96%; 95% CI: 91% to 99%) were performed by the selected provider on the given date. Thus, all 7 providers met inclusion criteria with ≥10 procedures performed in the last 12 months, with a PPV of 100% at the provider level. For the 4 surveillance procedures that were not confirmed by chart review, the reason was the absence of a procedure note. Thus, it remained unclear whether the procedure was performed or not.

## Discussion

We found that EHR data can be used for accurate identification of healthcare providers who meet patient- and provider-level inclusion criteria and thus provides a practical approach for implementation research. Starting with almost 2 million cystoscopy procedures and ~15,000 providers, we leveraged patient- and provider-level EHR data to identify 61 providers who met our predefined inclusion criteria.

Previous studies have investigated the use of EHRs for patient screening for recruitment as research participants [25]. For patient recruitment, EHRs have effectively reduced both the time and cost in the recruitment of patients for randomized clinical trials. As such, study timelines were consequently accelerated, and research investigators could focus their efforts on other aspects of the study [12,25]. Even with these described advantages for patient screening, EHR use in provider screening has not been described. Although EHR's have recently been reported as a major source of physician burnout [26], they also show great potential benefits for clinical and implementation science research studies. In this study, we extrapolated methods previously used for EHR-based patient screening to that of provider screening for study inclusion. Our study demonstrates the practicality of research based on the EHR for simple procedures or well described diseases. Both EHR vendors and coding authorities could be valuable stakeholders in the expansion and improvement of the quality of EHR based research.

Self-report and manual chart review can also be effective screening methods and could alternatively have been used in our study instead of the automated EHR approach. However, both alternative screening methods come with limitations (Fig 3). Employing the self-report

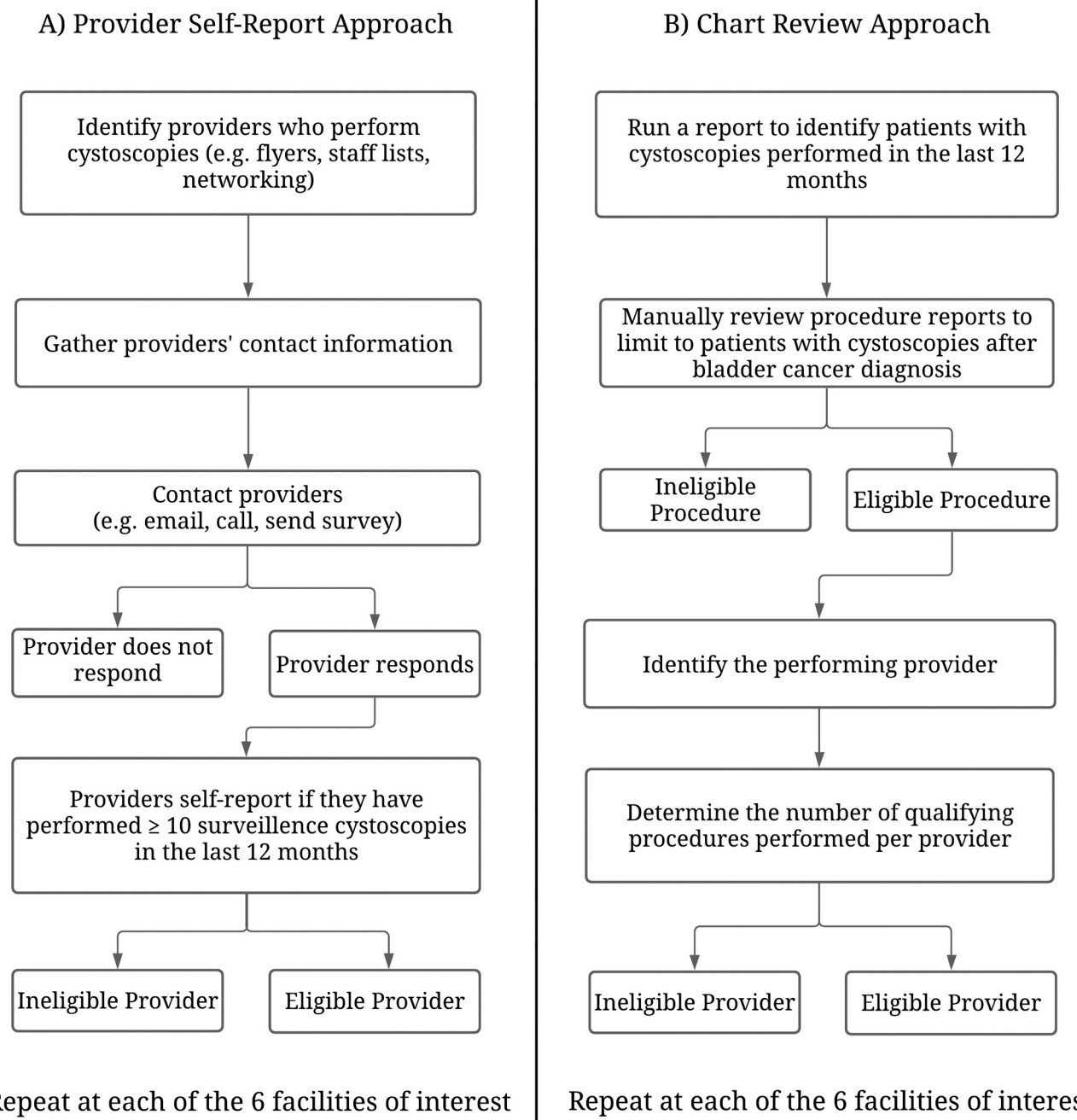

**A) Provider Self-Report Approach**

**B) Chart Review Approach**

Repeat at each of the 6 facilities of interest

Repeat at each of the 6 facilities of interest

**Fig 3.** Flow chart of alternative approaches for identifying eligible providers: A) Provider self-reported B) Manual chart review.

approach would have required us to identify and contact all currently practicing providers who perform cystoscopies from the six VHA facilities of interest. Then, these providers would have had to self-report how many surveillance procedures they performed in the previous 12 months–a method subject to recall bias [6]. As providers may perform many dozens or even hundreds of cystoscopy procedures per year on varying patients, memory decay is expected. The manual chart review approach would have been similarly cumbersome. With this approach, using clinic records at the specified VHA facilities, we would have had to identify

patients who recently underwent a cystoscopy procedure within the last 12 months. In addition, we would have needed to find a subset of those patients who had their cystoscopy for surveillance of their bladder cancer. Within that cohort, we could use this information to identify performing and eligible providers. (Fig 3). This approach would be time-consuming and subject to variation between chart reviewers/study coordinators [5]. These limitations posed by these known approaches are why an EHR facilitated approach for provider eligibility recruitment may be preferred [27,28].

An additional strength of EHR-based selection approach is its flexibility in evaluating granular inclusion criteria dictated by the study aim. In this study providers were eligible if they performed 10 procedures in the previous year regardless of how many unique patients were involved (Table 1). Additional patient or provider specific requirements could easily be incorporated into the coding process. A final strength of this approach is its generalizability. The current study used VHA EHR data. While not all healthcare systems have a vast EHR-based data resource [29], administrative codes and provider identifiers are collected in the majority of EHRs for billing purposes. As such, our approach can likely be translated to provider recruitment in other healthcare systems.

However, there are also limitations to the EHR approach that warrant discussion. One limitation to this approach is the misclassification of data elements due to coding errors. We observed 4 false-positive surveillance procedures. The reason was a missing procedure note. It was unclear whether the procedure happened or not. It is possible that providers forgot to enter the note or that a procedure was erroneously coded although it was never done. Administrative coding (e.g., ICD, CPT-4, HCPCS) is imperfect and miscoding or non-specific coding may limit their utility in certain research settings, including EHR facilitated provider recruitment. For example, codes may not accurately reflect a patient's underlying disease or what occurred in clinical practice. In urology specifically, there are certain diseases such as pyonephrosis and chronic testicular pain that have non-specific diagnosis codes [30,31]. Codes for these diseases may be clinically accurate but lack the granularity required to identify a specific subset of patients. In this present study of surveillance cystoscopy for bladder cancer patients, coding was sufficient to identify qualifying providers. However, structured data may not adequately or reliably capture other clinical domains as well and manual chart review alone or as a supplement of EHR based selection may be needed to overcome this limitation. A second limitation of this study is the somewhat limited validation given that there was no gold standard approach to identify surveillance cystoscopy or eligible providers. Thus, we were able to assess PPV but were unable to assess the sensitivity of the EHR based approach for identifying surveillance cystoscopy procedures or for providers performing surveillance cystoscopy. As such, our approach may have missed some qualifying procedures or providers. However, both misclassification and unknown sensitivity were less of a concern given that cystoscopy is a common procedure performed at the VHA. Thus, a few missed procedures even if misclassified would likely not have affected a provider's eligibility. However, we acknowledge that for less common procedures, sensitivity would be a more important metric as even a small amount of misclassification could make a provider erroneously eligible or ineligible.

In conclusion, the EHR-based screening model appreciably simplifies identification of eligible providers for research investigations, compared to alternative methods like manual chart review or self-report. Our EHR-based screening approach can likely be adapted for use in any healthcare system with an established EHR. Given the extensive array of information collected by healthcare systems in EHRs (diagnosis, medication, clinical notes, laboratory tests, consults, etc.), there are many ways research investigators can utilize this EHR-based screening approach to identify providers who meet defined inclusion criteria. We suggest researchers seriously consider EHR-based approaches to provider eligibility screening for their studies.

## Supporting information

**S1 Appendix. Cystoscopy procedures based on International Classification of Diseases, Ninth and Tenth Revision procedure codes and Common Procedural Terminology (CPT) codes.** Bladder Cancer diagnosis based on the International Classification of Diseases, Ninth and Tenth Revision diagnosis codes (ICD-9, ICD-10).
(DOCX)

## Acknowledgments

**Disclaimer.** Opinions expressed in this manuscript are those of the authors and do not constitute official positions of the US Federal Government or the Department of Veterans Affairs. The funding organizations had no role in the design and conduct of the study; collection, management, analysis, and interpretation of the data; preparation, review, or approval of the manuscript; and decision to submit the manuscript for publication.

## Author Contributions

**Conceptualization:** Chiamaka L. Okorie, Florian R. Schroeck, Kristine E. Lynch.

**Data curation:** Elise Gatsby, A. Aziz Ould Ismail, Kristine E. Lynch.

**Formal analysis:** Elise Gatsby, Kristine E. Lynch.

**Funding acquisition:** Florian R. Schroeck.

**Investigation:** Chiamaka L. Okorie, Florian R. Schroeck, A. Aziz Ould Ismail, Kristine E. Lynch.

**Methodology:** Chiamaka L. Okorie, Elise Gatsby, Florian R. Schroeck, A. Aziz Ould Ismail, Kristine E. Lynch.

**Project administration:** Chiamaka L. Okorie, Elise Gatsby, Florian R. Schroeck, Kristine E. Lynch.

**Resources:** Florian R. Schroeck.

**Software:** Elise Gatsby, A. Aziz Ould Ismail, Kristine E. Lynch.

**Supervision:** Florian R. Schroeck, Kristine E. Lynch.

**Validation:** Elise Gatsby, Florian R. Schroeck, Kristine E. Lynch.

**Visualization:** Chiamaka L. Okorie, Elise Gatsby, A. Aziz Ould Ismail.

**Writing – original draft:** Chiamaka L. Okorie.

**Writing – review & editing:** Chiamaka L. Okorie, Elise Gatsby, Florian R. Schroeck, A. Aziz Ould Ismail, Kristine E. Lynch.

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
