## [Decision Letter · Decision Letter 0]

11 Mar 2022

PONE-D-21-36556Using Electronic Health Records to Streamline Provider Recruitment for Implementation Science StudiesPLOS ONE

Dear Dr. Okorie,

Thank you for submitting your manuscript to PLOS ONE. After careful consideration, we feel that it has merit but does not fully meet PLOS ONE’s publication criteria as it currently stands. Therefore, we invite you to submit a revised version of the manuscript that addresses the points raised during the review process.

We look forward to receiving your revised manuscript.

Kind regards,

Beatrice Nardone

Academic Editor

PLOS ONE

Journal Requirements:

Reviewers' comments:

Reviewer's Responses to Questions

**Comments to the Author**

1. Is the manuscript technically sound, and do the data support the conclusions?

Reviewer #1: Yes

Reviewer #2: Yes

Reviewer #3: Yes

2. Has the statistical analysis been performed appropriately and rigorously? 

Reviewer #1: Yes

Reviewer #2: I Don't Know

Reviewer #3: Yes

3. Have the authors made all data underlying the findings in their manuscript fully available?

Reviewer #1: Yes

Reviewer #2: Yes

Reviewer #3: No

4. Is the manuscript presented in an intelligible fashion and written in standard English?

Reviewer #1: Yes

Reviewer #2: Yes

Reviewer #3: Yes

5. Review Comments to the Author

Reviewer #1: There is clear methodology in the method section and well description of the topic in the introduction part

result is very consistent with discussion part

I recommend the author to add some strength and limitation of the study

Reviewer #2: This is a well written study which is in line with the new trend in the researchers and what EMR providers promotes in their marketing.

I would suggest adding on the discussion that despite the fact that EMR is one of the major sources of physicians burnout, it has its own benefits such as the help in the research projects. I would also emphasize more on the coding errors and also the fact that coding systems have their own limitations and therefore using EMR based research may not be always ideal. In urology for example, there are several diseases which you may not be able to find an appropriate code for them as a diagnostic code, and there are also several codes for a single disease which can be misleading when it comes to conducting a research project on a disease or a procedure. I believe your study can be considered as an example of practicality of research based on EMR systems for simple procedures or some clearly defined and understood diseases at this stage, and both EMR companies and coding authorities should help with the improvement of the quality of these types of the researches which will eventually be the main source of clinical research in near future.

Reviewer #3: Okorie et al evaluated the effectiveness of using EHR to identify potential healthcare providers who had performed >=10 surveillance cystoscopy for bladder cancer patients within 12 months prior to the recruitment and were practicing at specified facilities of Veteran Affairs. They found records from EHR had similar performance with usual chart review with a positive predictive value of 96% for correctly identify surveillance cystoscopy procedures.

Overall, the study is rigorously conducted, and the paper provides new insights for finding eligible providers for multi-center clinical research.

Comments:

1. Practicing urologists for surveillance cystoscopy: The authors found sixty-one out of 1,005 (6%) current practicing providers that met the inclusion criteria. When limiting current practicing providers to urologists, a different picture might be shown. The VA healthcare provider website (https://www.accesstocare.va.gov/ourproviders ) lists 65, 59, and 46 physicians in surgery practicing at White River Junction, Salt Lake City, and Richard L. Roudebush (facilities listed in Acknowledgements). It is expected that the total number of urologists will be less. So, how is the ratio of sixty-one to all current practicing urologists at specified facilities?

2. Procedures and patients: despite the same statistics, performing the procedure ten times for one patient differs from that one time for ten patients. To further demonstrate the superiority of EHR-based provider selection, please provide a table detailing the number (median and interquartile range) of procedures and unique patients for selected provider groups listed in Figure 2.

6. PLOS authors have the option to publish the peer review history of their article (what does this mean?). If published, this will include your full peer review and any attached files.

Reviewer #1: **Yes: **Mohammedjud Hassen Ahmed

Reviewer #2: No

Reviewer #3: No

---

## [Author Response · Author response to Decision Letter 0]

6 Apr 2022

Reviewer #1: Mohammedjud Hassen Ahmed

Attests that our manuscript is technically sound, presented in an intelligible fashion with data that support the conclusions made using appropriate and rigorous statistical analysis. Dr. Hassen Ahmed also accepts our data sharing restriction. 

Author Response: Thank you very much.

Comment 1. 

There is clear methodology in the method section and well description of the topic in the introduction part. Result is very consistent with discussion part. 

Author Response: Thank you Dr. Hassen Ahmed.

Comment 2. 

I recommend the author to add some strength and limitation of the study 

Author Response: Thank you for this suggestion. We agree that an additional discussion of the strengths and limitations to the study is needed, and this was also suggested by the other two reviewers. We have provided an additional limitation and three study strengths. The additional statements with some supporting references for each added strength and limitation are below.

Strengths:

1) We acknowledge the practicality of EHR data as an avenue for major clinical research in the second paragraph of the discussion section. (page 8, line 292-294)

“Our study demonstrates the practicality of research based on the EHR for simple procedures or well described diseases. Both EHR vendors and coding authorities could be valuable stakeholders in the expansion and improvement of the quality of EHR based research.”

2) We demonstrated the utility of EHR driven selection by showing that EHR helps distinguish between patient level and procedure level data. Statement below to be found in the fourth paragraph of the discussion section (page 8, line 312-315). See result section for Table 1.

“An additional strength of EHR-based selection approach is its flexibility in evaluating granular inclusion criteria dictated by the study aim. In this study providers were eligible if they performed 10 procedures in the previous year regardless of how many unique patients were involved (Table 1). Additional patient or provider specific requirements could easily be incorporated into the coding process.”

3) We discussed the generalizability of this EHR-based approach. Statement below to be found in the fourth paragraph of the discussion section (page 8, line 315-319).

“A final strength of this approach is its generalizability. The current study used VHA EHR data. While not all healthcare systems have a vast EHR-based data resource, [29] administrative codes and provider identifiers are collected in the majority of EHRs for billing purposes. As such, our approach can likely be translated to provider recruitment in other healthcare systems.”

29. Velarde KE, Romesser JM, Johnson MR, Clegg DO, Efimova O, Oostema SJ, et

al. An initiative using informatics to facilitate clinical research planning and recruitment in the VA health care system. Contemporary Clinical Trials Communications. 2018;11: 107–112. doi:10.1016/j.conctc.2018.07.001

Limitations:

1) We added a statement on Coding errors as a limitation to EHR based studies in the fifth paragraph of the discussion section including a reference on coding errors in urology. (page 9, line 331-340)

“Administrative coding (e.g., ICD, CPT-4, HCPCS) is imperfect and miscoding or non-specific coding may limit their utility in certain research settings, including EHR facilitated provider recruitment. For example, codes may not accurately reflect a patient’s underlying disease or what occurred in clinical practice. In urology specifically, there are certain diseases such as pyonephrosis and chronic testicular pain that have non-specific diagnosis codes. [30,31] Codes for these diseases may be clinically accurate but lack the granularity required to identify a specific subset of patients. In this present study of surveillance cystoscopy for bladder cancer patients, coding was sufficient to identify qualifying providers. However, structured data may not adequately or reliably capture other clinical domains as well and manual chart review alone or as a supplement of EHR based selection may be needed to overcome this limitation.” 

30. Ballaro A, Oliver S, Emberton M. Do we do what they say we do? Coding errors in urology. BJU International. 2000;85: 389–391. doi:10.1046/J.1464-410X.2000.00471.X

31. Khwaja HA, Syed H, Cranston DW. Coding errors: a comparative analysis of hospital and prospectively collected departmental data. BJU international. 2002;89: 178–180. doi:10.1046/J.1464-4096.2001.01428.X

Reviewer #2:

Has the statistical analysis been performed appropriately and rigorously?

Reviewer #2: I Don't Know

Author Response: We are unsure what specific statistical analysis Reviewer #2 is uncertain about. We are happy to elaborate on our statistical analysis with further comments. To the best of our current knowledge, all statistical analyses were performed appropriately and rigorously.

Comment 1. 

This is a well written study which is in line with the new trend in the researchers and what EMR providers promotes in their marketing.

Author Response: We thank the reviewer for their time and helpful comments. 

Comment 2. 

I would suggest adding on the discussion that despite the fact that EMR is one of the major sources of physician’s burnout, it has its own benefits such as the help in the research projects. 

Author Response: Thank you for this suggestion. We agree with your comment and have incorporated your suggestion to the second paragraph of the discussion section (page 8 line 289-290). The added statement is also below including a reference on EHR contributing to physician burnout.

“Although EHR’s have recently been reported as a major source of physician burnout,[26] they also show great potential benefits for clinical and implementation science research studies.”

26. Babbott S, Manwell LB, Brown R, Montague E, Williams E, Schwartz M, et al. Electronic medical records and physician stress in primary care: Results from the MEMO Study. Journal of the American Medical Informatics Association. 2014;21. doi:10.1136/amiajnl-2013-001875

Comment 3. 

I would also emphasize more on the coding errors and the fact that coding systems have their own limitations and therefore using EMR based research may not be always ideal. In urology for example, there are several diseases which you may not be able to find an appropriate code for them as a diagnostic code, and there are also several codes for a single disease which can be misleading when it comes to conducting a research project on a disease or a procedure. 

Author Response: Thank you for this comment. We have added a description of this limitation to the fifth paragraph of the discussion section (page 9, line 331-340). The added statement is also below including a reference on coding errors in urology. 

“Administrative coding (e.g., ICD, CPT-4, HCPCS) is imperfect and miscoding or non-specific coding may limit their utility in certain research settings, including EHR facilitated provider recruitment. For example, codes may not accurately reflect a patient’s underlying disease or what occurred in clinical practice. In urology specifically, there are certain diseases such as pyonephrosis and chronic testicular pain that have non-specific diagnosis codes. [30,31] Codes for these diseases may be clinically accurate but lack the granularity required to identify a specific subset of patients. In this present study of surveillance cystoscopy for bladder cancer patients, coding was sufficient to identify qualifying providers. However, structured data may not adequately or reliably capture other clinical domains as well and manual chart review alone or as a supplement of EHR based selection may be needed to overcome this limitation.” 

30. Ballaro A, Oliver S, Emberton M. Do we do what they say we do? Coding errors in urology. BJU International. 2000;85: 389–391. doi:10.1046/J.1464-410X.2000.00471.X

31. Khwaja HA, Syed H, Cranston DW. Coding errors: a comparative analysis of hospital and prospectively collected departmental data. BJU international. 2002;89: 178–180. doi:10.1046/J.1464-4096.2001.01428.X

Comment 4. 

I believe your study can be considered as an example of practicality of research based on EMR systems for simple procedures or some clearly defined and understood diseases at this stage, and both EMR companies and coding authorities should help with the improvement of the quality of these types of the research which will eventually be the main source of clinical research in near future. 

Author Response: Thank you for this comment. We appreciated this comment and have included it in the discussion in the second paragraph (page 8/Line 292-294). The summarized statement is below. 

“Our study demonstrates the practicality of research based on the EHR for simple procedures or well described diseases. Both EHR vendors and coding authorities could be valuable stakeholders in the expansion and improvement of the quality of EHR based research.”

Reviewer #3: 

Have the authors made all data underlying the findings in their manuscript fully available?

Reviewer #3: No

Author Response: We unfortunately cannot make the data publicly available as they contain potentially identifying and sensitive patient information. Data in the Department of Veterans Affairs Corporate Data Warehouse are collected for clinical purposes as part of the patient medical record. They contain potentially identifying and sensitive patient information and, therefore, cannot be shared. They can be accessed by any VA researcher through the Institutional Review Board process. Interested researchers can direct data access requests to the director of the Veteran's IRB of Northern New England, 215 N Main Street, White River Junction, VT 05009, phone 802-295-9363, email: vhawrjresearchtask@va.gov. 

Comment 1. 

Okorie et al evaluated the effectiveness of using EHR to identify potential healthcare providers who had performed >=10 surveillance cystoscopy for bladder cancer patients within 12 months prior to the recruitment and were practicing at specified facilities of Veteran Affairs. They found records from EHR had similar performance with usual chart review with a positive predictive value of 96% for correctly identify surveillance cystoscopy procedures.

Overall, the study is rigorously conducted, and the paper provides new insights for finding eligible providers for multi-center clinical research.

Author Response: Thank you for your time and helpful comments.

Comment 2. 

Practicing urologists for surveillance cystoscopy: The authors found sixty-one out of 1,005 (6%) current practicing providers that met the inclusion criteria. When limiting current practicing providers to urologists, a different picture might be shown. The VA healthcare provider website (https://www.accesstocare.va.gov/ourproviders) lists 65, 59, and 46 physicians in surgery practicing at White River Junction, Salt Lake City, and Richard L. Roudebush (facilities listed in Acknowledgements). It is expected that the total number of urologists will be less. So, how is the ratio of sixty-one to all current practicing urologists at specified facilities? 

Author Response: Thank you for bringing up this concern. We are grateful for the level of investigation put in by this reviewer. We understand how 61 can exceed expectations, but it is an accurate representation of the sites due to the following:

• We assessed 6 total facilities, of which only three where highlighted in the Acknowledgements.

• In addition to attending urologists, residents, and Advance Practice Providers (NP/PA) were considered for inclusion and considered currently practicing if at least one of their qualifying procedures was performed at one of the 6 study sites. Given the high turnover in these positions, residents in particular, these providers add substantially to the reported number. 

• We anticipate readers having the same concern as this reviewer and have added language in text in the methods section: eligibility criteria (page 4, line 148-151) to clarify that providers included but were not limited to attending urologists. The added statement is below.

“Attending urologists, residents, and Advance Practice Providers (NP/PA) were all considered for inclusion and considered currently practicing if at least one of their qualifying procedures was performed at one of the 6 study sites.”

• Lastly, while investigating your comment, we noticed a typographic error in Figure 2 that may have caused confusion. Figure 2 previously showed there were 1,005 providers who performed at least 1 cystoscopy were currently practicing at one of the 6 stations. It now correctly states that there were 1,005 providers who performed at least 1 cystoscopy at one of the 6 stations at any point in time (1999-2020). 

Comment 3. 

Procedures and patients: despite the same statistics, performing the procedure ten times for one patient differs from that one time for ten patients. To further demonstrate the superiority of EHR-based provider selection, please provide a table detailing the number (median and interquartile range) of procedures and unique patients for selected provider groups listed in Figure 2.

Author Response: We agree with the reviewer’s assessment that performing a cystoscopy ten times on the same patient is quite different than one time for ten patients. Although this distinction did not matter for our study inclusion criteria, it could for a different study. We add this as an additional strength of EHR-based provider selection in the fourth paragraph of the discussion section (page 8, line 312-315). The added statement is below. 

“An additional strength of EHR-based selection approach is its flexibility in evaluating granular inclusion criteria dictated by the study aim. In this study providers were eligible if they performed 10 procedures in the previous year regardless of how many unique patients were involved (Table 1). Additional patient or provider specific requirements could easily be incorporated into the coding process.”

Additionally, as requested by the reviewer, we have provided a table (Table 1) detailing the number (median and interquartile range) of procedures and unique patients for selected provider groups in the results section under provider level, as it further demonstrates the utility of EHR driven selection. The added statement is below in the result section: provider level (page 4, line 232-235).

“These 61 providers were considered eligible for recruitment (Figure 2) with an average of 31.4 patients seen by these eligible providers and an average of 34.1 procedures performed. Below we provide a table detailing the number (median and interquartile range) of procedures and unique patients for each selected provider groups before final eligibility criteria were met. (Table 1)”

Table 1: Detailing the minimum, median and maximum number of procedures and unique patients for selected provider groups with averages and SD, including the specific procedure timeline.

---

## [Decision Letter · Decision Letter 1]

19 Apr 2022

Using Electronic Health Records to Streamline Provider Recruitment for Implementation Science Studies

PONE-D-21-36556R1

Dear Dr.  Okorie,

We’re pleased to inform you that your manuscript has been judged scientifically suitable for publication and will be formally accepted for publication once it meets all outstanding technical requirements.

Kind regards,

Beatrice Nardone

Academic Editor

PLOS ONE

Additional Editor Comments (optional):

Reviewers' comments:

Reviewer's Responses to Questions

**Comments to the Author**

1. If the authors have adequately addressed your comments raised in a previous round of review and you feel that this manuscript is now acceptable for publication, you may indicate that here to bypass the “Comments to the Author” section, enter your conflict of interest statement in the “Confidential to Editor” section, and submit your "Accept" recommendation.

Reviewer #2: All comments have been addressed

Reviewer #3: All comments have been addressed

2. Is the manuscript technically sound, and do the data support the conclusions?

Reviewer #2: Yes

Reviewer #3: (No Response)

3. Has the statistical analysis been performed appropriately and rigorously? 

Reviewer #2: (No Response)

Reviewer #3: (No Response)

4. Have the authors made all data underlying the findings in their manuscript fully available?

Reviewer #2: Yes

Reviewer #3: (No Response)

5. Is the manuscript presented in an intelligible fashion and written in standard English?

Reviewer #2: Yes

Reviewer #3: (No Response)

6. Review Comments to the Author

Reviewer #2: Authors have addressed all of my comments. With the statistical analysis aspect of the manuscript, I would not consider myself competent enough to investigate the details and the editor will decide.

Reviewer #3: (No Response)

7. PLOS authors have the option to publish the peer review history of their article (what does this mean?). If published, this will include your full peer review and any attached files.

Reviewer #2: No

Reviewer #3: No

---

## [Editor Report · Acceptance letter]

6 May 2022

PONE-D-21-36556R1 

Using Electronic Health Records to Streamline Provider Recruitment for Implementation Science Studies. 

Dear Dr. Okorie:

I'm pleased to inform you that your manuscript has been deemed suitable for publication in PLOS ONE. Congratulations! Your manuscript is now with our production department. 

Kind regards, 

on behalf of

Dr. Beatrice Nardone 

Academic Editor

PLOS ONE